# Mechanisms of Health Improvement by *Lactiplantibacillus plantarum* Based on Animal and Human Trials: A Review

Yu Hao [1,2,3], Jianli Li [1,2,3], Jicheng Wang [1,2,3,*] and Yongfu Chen [1,2,3,*]

1   Key Laboratory of Dairy Biotechnology and Engineering, Ministry of Education, Inner Mongolia Agricultural University, Hohhot 010018, China; haoyu6878@163.com (Y.H.); lijianli107@163.com (J.L.)
2   Key Laboratory of Dairy Products Processing, Ministry of Agriculture and Rural Affairs, Inner Mongolia Agricultural University, Hohhot 010018, China
3   Inner Mongolia Key Laboratory of Dairy Biotechnology and Engineering, Inner Mongolia Agricultural University, Hohhot 010018, China
*   Correspondence: imwjc@163.com (J.W.); nmgyfchen@126.com (Y.C.)

**Abstract:** *Lactiplantibacillus plantarum* is a candidate probiotic that has been included in the list of recommended biological agents for certification by the European Food Safety Authority. It has been found to be widely present in acidic-gruel, yogurt, cheese, kefir, kimchi, and so on. In this article, we have reviewed both preclinical and human studies related to the health promoting effects of *L. plantarum* that have been published for the past decade. We found that *L. plantarum* could significantly improve intestinal function, oral as well as skin health, promote neuro as well as immune regulation, and be effective against metabolic diseases, etc. *L. plantarum* primarily enters the body through the oral cavity and acts on the gastrointestinal tract to effectively improve the intestinal flora. It can affect the female reproductive endocrine system through interaction with estrogen, androgen, insulin, and other hormones, as well as improve the body's allergic reaction and immunity by regulating Th1/Th2 response. Several prior reports also suggest that this Gram-positive bacterium can promote production and secretion of key neurotransmitters and neural activators in the intestinal tract by regulating the intestinal flora by directly or indirectly affecting the gut–brain axis through modulation of vagus nerve, cytokines, and microbial metabolites, thus relieving stress and anxiety symptoms in adults. This review is the first report describing the health promoting effects of *L. plantarum*, with the aim of providing a theoretical basis for the development of various beneficial applications of *L. plantarum*.

**Keywords:** *L. plantarum*; intestinal microorganism; nervous system; metabolic capability; oral; skin





## 1. Introduction

Probiotics is the general term used for active beneficial microorganisms that are colonized in the human intestinal tract and reproductive system. They have been found to exhibit significant health effects to improve the ecological balance of the host and play a beneficial role in the host [1]. They can inhibit the systemic invasion of various pathogens through the gastrointestinal mucosa or oral cavity. They have been found to display effective preventive, as well as therapeutic effects, on the different infectious pathogens and non-communicable diseases. They can also regulate host immune response, thus contributing to the fight against immune dysfunction. In addition to being used for immune-related diseases, probiotics have been reported to effectively treat diarrhea, bacterial vaginosis, urinary tract infection, dental pulp disease, diabetes, cancer, as well as other diseases. They have also been reported to enhance the tolerance of intestinal mucosa to various antibiotics and alleviate the symptoms of lactose intolerance [2]. *Lactiplantibacillus plantarum* is an important candidate strain for probiotics which is widely distributed in the nature. Several previous studies have explored the probiotic properties of *L. plantarum* in a more comprehensive manner. The potential health

promoting effects of *L. plantarum* on the host have been demonstrated by its anti-fungal as well as anti-viral activities, bacteriocin production, and function as a exopolysaccharide.

The potential commercial prospects of *L. plantarum* have generated wide interest in recent years. Importantly, researchers are interested in strategies to further improve the various reported health benefits of *L. plantarum*. The improvement from *L. plantarum* on intestinal flora is an important research subject, which can be potentially extended to study nervous system and metabolic diseases associated with intestinal flora. Concurrently, skin, oral, and other fields are also important forms of *L. plantarum*. In summary, *L. plantarum* has shown significant effects on human health. Therefore, we selected the preclinical and human trials around *L. plantarum* in the past decade to compile a comprehensive overview.

## 2. Physiological Characteristics of *L. plantarum*

*L. plantarum* is widely distributed in nature [3] and belongs to the genus Lactobacilli and Gram-positive bacteria. It only produces lactic acid during the fermentation process and is a typical facultative anaerobe. It possesses a strong ability to ferment carbohydrates, is salt-tolerant, and can exhibit synergistic effects with other lactic acid bacteria (LAB). Under the electron microscope, it appears as long rod-shaped, with uniform thickness and blunt end, without flagella and spores (Figure 1). This bacterium can grow under a temperature of 15~45 °C, with optimal growth temperature being 30~37 °C [4,5], with a required pH of around 6.5. It can be isolated from the digestive system of humans and mammals [6], the urogenital tract [7], and also grows well on the natural fermented foods such as sour porridge, pickled cabbage, sour horse milk, and kefir [8,9], which are recognized as probiotics candidates.

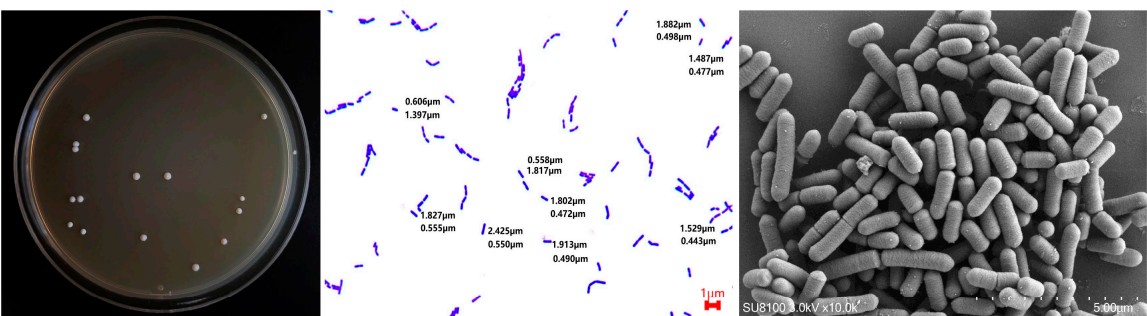

**Figure 1.** Microscopic morphology of *L. plantarum* P-8 [10].

## 3. Market Prospect of *L. plantarum*

*L. plantarum* has been widely used for the fermentation of fruits, vegetable juices [11], dairy products [12], cereal-based products [13], and meat products [14]. In addition, it is also extensively employed as a biological preservative in the food industry [15] and has several applications in the medical fields, such as in the treatment of conditions of oral cavity, skin, veterinary, and other fields. *L. plantarum* can also play a vital role in animal husbandry, environmental ecology, cosmetics, and other industries. We have compiled the reported applications of different strains of *L. plantarum* in Table 1.

**Table 1.** Physiological characteristics and functions of the commercial probiotics.

| Strain | Source | Physiological Function | Experimental Object | Reference |
|--------|--------|------------------------|---------------------|-----------|
| *L. plantarum* CJLP133 | Kimchi | Antiallergy, regulating Th1/Th2 balance | Animal (mice) trials | [16] |
| *L. plantarum* CJLP243 | | | | [17] |
| *L. plantarum* YIT0132 | Fermented food | | Human trials | [18] |
| *L. plantarum* L-137 | A traditional fermented food produced in the Philippines | Antiallergy, regulating Th1/Th2 balance; degraded pesticide | Animal (mice) trials | [19] |

**Table 1.** *Cont.*

| Strain | Source | Physiological Function | Experimental Object | Reference |
|---|---|---|---|---|
| *L. plantarum* LRCC5310 | Kimchi | Produced EPS, resists virus, and relieves intestinal problems such as diarrhea caused by rotavirus | Animal (mice) trials and human trials | [20,21] |
| *L. plantarum* NCU116 | Pickled vegetables | Ameliorated Type 2 diabetes; reduce cholesterol | Animal (mice) (rat) trials | [22–24] |
| *L. plantarum* VSG3 | The gut of healthy rohu (Labeo rohita) | Enhanced immunity and anti-streptococcal infection ability | Animal (fish) trials | [25] |
| *L. plantarum* DR7 | Bovine milk | Alleviated the symptoms of upper respiratory tract infection; regulates immunity | Human trials | [26] |
| *L. plantarum* PFM105 | The rectum of a healthy sow | Improved the intestinal flora, increased the number of beneficial bacteria in the intestine, and reduced the number of conditional pathogenic bacteria; attenuated resistance genes and antibiotic residues | Animal (weaning piglet) trials | [27] |
| *L. plantarum* 299v | Dehydrated fermented milk | Improved intestinal flora and female reproductive system; enhanced the memory | Human trials | [28] |
| *L. plantarum* I-UL4 | Malaysian Tempeh | Reduced cholesterol and increases the number of intestinal beneficial bacteria | Animal (post weaning rats) trials | [29] |
| *L. plantarum* P9 | Traditional fermented food | Improved the degradation rate of pesticides and the stability of intestinal flora | Animal (mice) trials | [30] |
| *L. plantarum* CCFM8661 | Kimchi | Modulated bile acid enterohepatic circulation and promotes heavy metal excretion | Animal (mice) trials | [31] |
| *L. plantarum* CCFM8610 | Kimchi | Alleviated irritable bowel syndrome and prevents gut microbiota dysbiosis | Animal (mice) trials | [32,33] |
| *L. plantarum* LP33 | Yogurt, Xinjiang, China | Promoted heavy metal excretion, regulates oxidative stress, and alleviates heavy metal toxicity in the tissues | Animal (rat) trials | [34] |
| *L. plantarum* TWK10 | Taiwan pickled vegetables | Improved exercise performance and increased muscle mass | Animal (mice) (rat) trials and human trials | [35–37] |
| *L. plantarum* PS128 | Traditional fermented food–Fu-Tsai | Improved the depressive symptoms and sleep quality of insomniacs and in allay autism spectrum disorder (ASD) | Human trials | [38–40] |
| *L. plantarum* ST-III | Homemade pickles in China | Improved social behavior in a male mouse model of ASD and contribute to more balanced intestinal homeostasis. | Animal (mice) trials | [41] |
| *L. plantarum* P17630 | Healthy vagina | Treated or alleviated letrozole-induced polycystic ovarian syndrome | Human trials | [42] |
| *L. plantarum* Q180 | The feces of adults | Reduced cholesterol | Human trials | [43] |
| *L. plantarum* CJLP55 | Kimchi | Improved the count and grading of acne lesions, increase hydration | Human trials | [44] |
| *L. plantarum* CCFM8724 | Fermented milk wine | Reduced the number of oral pathogens and altered the oral microbiota in children with dental caries. | Human trials | [45] |
| *L. plantarum* GMNL6 | - | Reduced the erythema and melanin, improve skin condition | Human trials | [46] |
| *L. plantarum* P8 | Natural fermented yoghurt | Improve intestinal flora, reduce cholesterol, regulate immunity and the nervous system | Animal (mice) (rat) trials and human trials | [26,47–49] |
| *L. plantarum* HY7714 | The breast milk of healthy women | Improved skin health and regulate permeability | Human trials | [50] |

## 4. *L. plantarum* with Intestinal Flora

### 4.1. Improvement in Intestinal Flora

The host intestinal flora represents a dense and competitive microbial ecosystem, which provides a group of core metabolic activities and immune stimulating molecules. For a long time, probiotics have been mainly introduced into the human body by mouth to produce the different beneficial effects, and the gastrointestinal tract has been identified as their primary target (Figure 2). Therefore, it is obvious that probiotics have a significant effect on the regulation of intestinal flora. The intestinal flora is a complex ecosystem composed of the various microorganisms. They jointly contribute to maintenance of the nutritional health, metabolism, tissue development, and immune maturation in the host [51]. In recent years, *L. plantarum* has been reported to promote the balance of host intestinal flora [52], and an indirect effect on body health has been found. For instance, in a human trial including the elderly, the middle-aged, and the young, *L. plantarum* P-8 was observed to significantly improve the structure of intestinal flora, which could be beneficial to the human health. Moreover, after administration of *L. plantarum* P-8 ($6 \times 10^{10}$ CFU/d) to the young, middle, and elderly people of three different age groups for 4 weeks, the number of *Bifidobacterium* and other beneficial bacteria was found to be increased. In addition, the number of Desulfovibrio and other conditional pathogenic bacteria decreased and the gastrointestinal microenvironment was markedly improved [47].

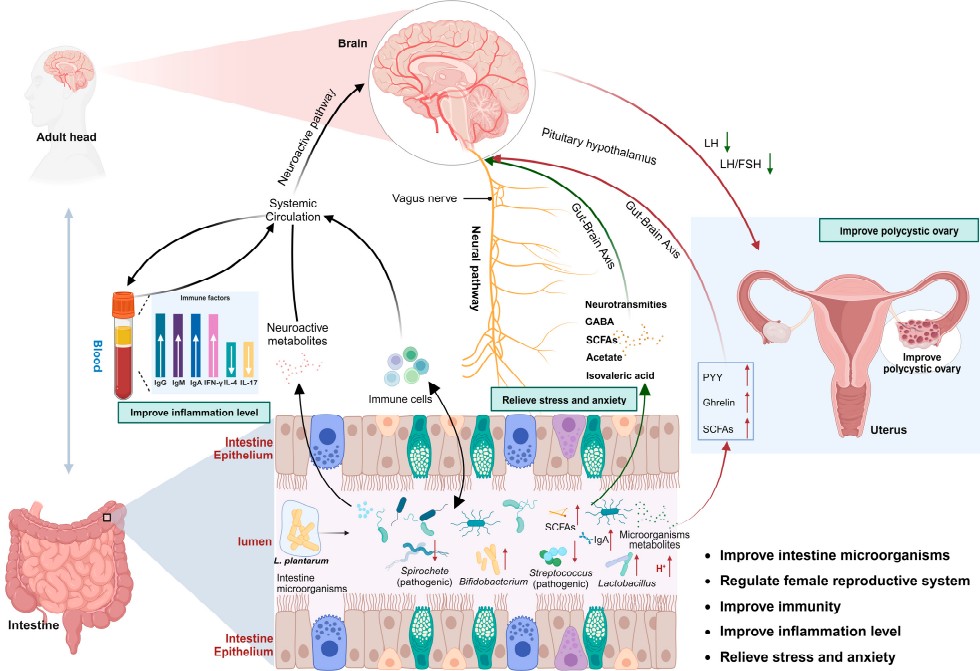

**Figure 2.** Interaction mechanism of *L. plantarum* with human intestinal flora and nervous system. The mode of action of exogenous *L. plantarum* on intestinal flora after ingestion (green arrow); Combined with the brain–gut axis effect, the potential impact on the nerves, stress, and brain mechanism; (Black arrow) Regulation of the level of inflammatory factors in the blood; (red arrow) effect on the female reproductive system and hormone production [53,54].

Moreover, several studies have shown that *L. plantarum* can significantly improve the intestinal flora, and has been reported to be useful for diarrhea, constipation, IBD (inflammatory bowel disease), as well as IBS (irritable bowel syndrome) caused by imbalance in the intestinal flora. *L. plantarum* was found to decrease the levels of pro-inflammatory cytokines, i.e., IL-17A, IL-17F, IL-6, IL-22, and TNF-$\alpha$, but increase that of anti-inflammatory cytokines, i.e., TGF-$\beta$, IL-10 [53]. *L. plantarum* ST-III intervention could effectively alter the intestinal microbiota structure, especially at the genus level, and significantly ele-

vate the proportion of *Sutterella*, *Pediococcus*, *Proteus*, *Parabacteroides*, *Prevotella*, as well as *Desulfovibrio*. Correlation analysis further revealed that the specific compositional features of intestinal microbiota were strongly related to the concentration of the cecal short-chain fatty acids (SCFAs) [41]. For instance, in a prior clinical trial, *L. plantarum* was found to significantly alleviate the symptoms of volunteers affected with IBS-D-associated diarrhea and improved their quality of life, stabilized the composition of enteric microorganisms, and increased the abundance of the beneficial bacterium [55]. *L. plantarum* 299v can also play a similar role because intake of *L. plantarum* 299v was observed to be beneficial for patients affected with IBS and could normalize the feces as well as relieve abdominal pain, thereby significantly improving the quality of life of IBS patients. In addition, *L. plantarum* 299v can effectively prevent Clostridium difficile-associated diarrhea in patients receiving antibiotic treatment [56,57]. In another study, DSS-induced mouse colitis model was established and mice were orally administered with *L. plantarum* (C2, C3, and P-8 $10^{10}$ CFU/mL, 30 d) to investigate its potential effects on colitis. The results indicated that *L. plantarum* substantially improved dysbiosis and enhanced the abundance of beneficial bacteria related to SCFAs production. Similarly, *L. plantarum* PFM105 could effectively reduce the high mortality, diarrheal rates, and intestinal inflammation caused by weaning of the piglets, thus leading to the promotion of the growth, development, and improvement in the structure of intestinal flora [27].

### 4.2. Intestinal Flora Mediates Female Reproductive Micro-Ecological System

The intestinal microflora plays an important role in the regulation of reproductive health, which mainly affects the female reproductive endocrine system through interaction with estrogen/androgen, insulin, and other hormones [58]. At present, scientific results have proved the various benefits of probiotics on the reproductive health outcomes. Most of the clinical trials related to management of the female reproductive disorders have been carried out by using oral probiotics [59]. In microbial reproductive disorders, probiotics should be transferred to the colonization sites of the microbial disorders (Figure 2), such as vaginal (vaginosis), endometrial (endometritis), or breast (mastitis). This metastasis could be achieved through physical ascending pathway, blood flow pathway, or lymph node metastasis [54].

In preclinical trials, *L. plantarum* CCFM1019 (CCFM1019) was used to treat or alleviate letrozole-induced polycystic ovarian syndrome (PCOS). The probiotics CCFM1019 could effectively attenuate the pathological changes in the ovaries and restore the levels of both testosterone and luteinizing hormone. In addition, alteration in gut microbial diversity and enrichment of the SCFAs producers *Lachnospira* and *Ruminococcus*_2 were observed after CCFM1019 intervention. It was noted that rats treated with CCFM1019 exhibited higher butyrate and polypeptide YY levels in comparison to letrozole-treatment, possibly due to the regulation of G protein-coupled receptor 41 expression. The investigators hypothesized that the brain–gut axis-dependent alleviation of letrozole-induced PCOS symptoms in rats might have been achieved through the modulation of butyrate metabolism [60]. Moreover, administration of *L. plantarum* P17630 capsules ($5 \times 10^9$ CFU/d) for 3 consecutive months for women with a history of recurrent vulvovaginal candidiasis [61] was reported to be superior upon comparison to placebo in reducing the various symptoms (redness and swelling) related to the recurrent vulvovaginal candidiasis and effectively improved the vaginal colonization of LAB. The swelling intensity of patients was observed to be significantly decreased on the 45th and 90th days. The application of this oral product could successfully prevent the attack of vulvovaginal candidiasis. Accordingly, compared with the long-term effect of oral probiotics, vaginal application of probiotics can achieve direct, faster, and targeted colonization to rapidly restore altered vaginal microflora. In the study of probiotics improving polycystic ovary syndrome, the mechanism of this effect may be related to the levels of luteinizing hormone (LH) and LH/follicle-stimulating hormone (LH/FSH) increased significantly, which affects the secretion of sex hormones in the pituitary hypothalamus through the gut–brain axis, thereby producing effects [42].

## 5. *L. plantarum* with Human Metabolism

### 5.1. Reducing Cholesterol

At present, incidences of various chronic metabolic diseases, including hyperlipidemia, hypertension, hyperglycemia, and hyperuricemia, are increasing rapidly worldwide. A number of young people are also being affected by such metabolic disorders. Hyperlipidemia is also an important risk factor for hypertension, impaired glucose tolerance, and diabetes. Here, we have discussed the potential of probiotics in affecting hyperlipidemia in detail [62].

Cholesterol is an essential nutrient for the human body, but its excessive intake can cause hyperlipidemia and complications. Several prior reports have confirmed that probiotics can reduce the medium as well as serum cholesterol, and one of the mechanisms postulated is that these probiotics can attenuate cholesterol levels primarily through assimilating and restraining the activity of cholesterol synthase (3-hydroxy-3-methyl-glutaryl-CoA reductase). It has been established that, as a result of the function of bile salt hydrolase (BSH), probiotics can promote the co-precipitation of the hydrolyzed bile salts in the small intestine with cholesterol in food, and thus can reduce the absorption of cholesterol by the body (Figure 3), thereby increasing the excretion of feces out of the body [63].

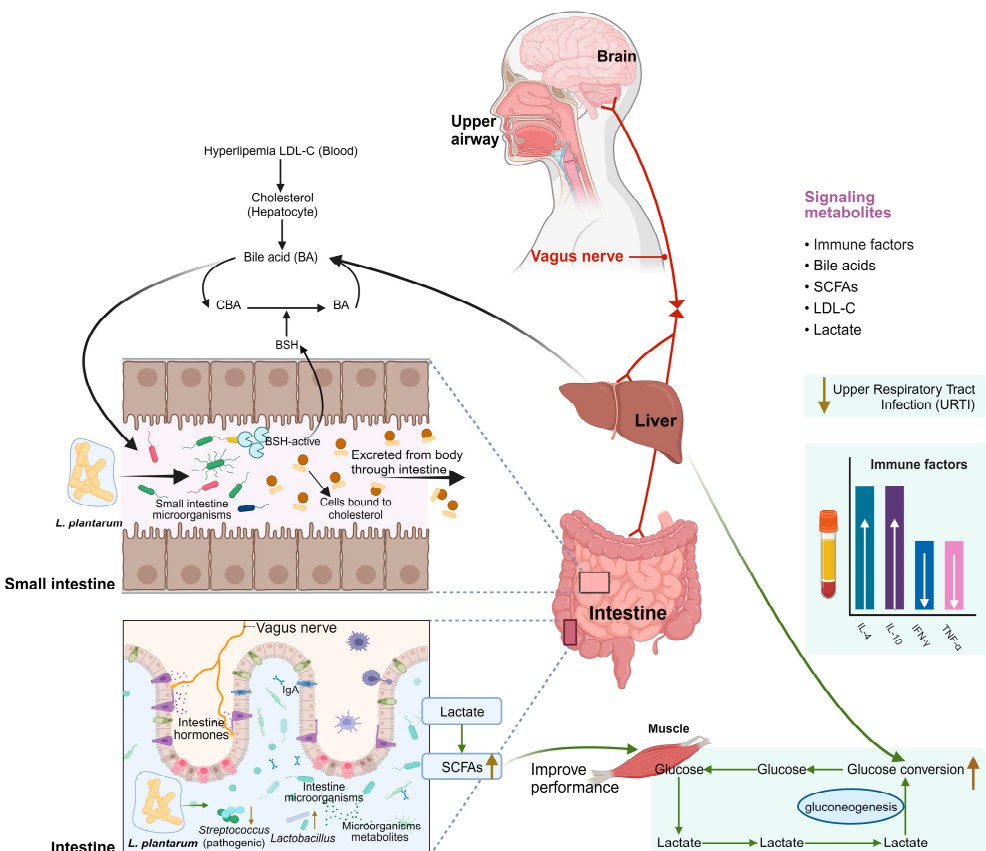

**Figure 3.** Interaction mechanism between *L. plantarum* and human metabolism and immune function. (black arrow) The mechanism of action of exogenous *L. plantarum* on cholesterol after food intake; (green arrow) Effects on the human motor performance, including the regulation of levels of some immune factor in the blood [63].

Several preliminary preclinical investigations have been conducted by different investigators about the potential of *L. plantarum* in improving hyperlipidemia. For example, Park Y E et al. investigated the effect of *L. plantarum* Q180 (LPQ180) on postprandial lipid metabolism and the intestinal microbiome environment from a clinical perspective. A double-blind, randomized, placebo-controlled study was conducted including 70 participants of both sexes, 20 years of age and older. The results show that treatment

with LPQ180 for 12 weeks ($4.0 \times 10^9$ colony-forming units/d) significantly decreased LDL-cholesterol [43]. Furthermore, in another parallel, double-blind, placebo-controlled, randomized pilot study, the participants were administered $4 \times 10^9$ CFU encapsulated *L. plantarum* ECGC13110402 (active; n = 8) over 6 weeks. Thereafter, the fasting blood samples were collected for analyses of blood lipid, liver function, and vitamin D. The findings suggested that *L. plantarum* ECGC13110402 could significantly as well as safely improve the lipid profiles in dyslipidaemic individuals [64]. In addition, when the mixed capsules containing *L. plantarum* (CECT7527, CECT7528, and CECT7529) ($12 \times 10^9$ CFU/d) were administered for 12 weeks, the plasma total cholesterol (TC) level in the *L. plantarum* group was significantly reduced by 13.6%. It was noted that in the high initial value group, the contents of TC, low-density lipoprotein cholesterol (LDL-C), and oxidized LDL-C decreased in *L. plantarum* group after 12 weeks. However, in the low initial value group, the *L. plantarum* group showed decreased TC only after 12 weeks. Therefore, the biological functions of *L. plantarum* (CECT7527, CECT7528, and CECT7529) can display significant therapeutic effects on patients with higher cholesterol levels [65].

Interestingly, prior studies have also indicated that the contents of total cholesterol, triglyceride, and low-density lipoprotein cholesterol in the serum and liver of rats with hyperlipidemia were significantly reduced after they were fed with *L. plantarum* P-8 ($1.0 \times 10^9$ CFU/d) for 28 days, which exhibited a substantial lipid-lowering effect [66]. Male mice were fed with a high-fat diet and orally administered with *L. plantarum* Q16 at a dose of $10^9$ CFU/mL for 8 weeks; subsequently, they effectively showed an improved serum and hepatic lipid profile and were protected against HFD-induced nonalcoholic fatty liver disease (NAFLD) by virtue of their improved hepatic profile as well as a balanced colonic microbiota composition [67]. C57BL/6N mice, which were fed HFD with *L. plantarum* 104 for 8 weeks, showed a significant decrease in the body weight, liver index, colon length, as well as TC, TG, LDL, TNFα, and lipopolysaccharide (LPS) levels. Moreover, *L. plantarum* improved the amelioration HFD induced hyperlipidemia by positively regulating intestinal microorganisms [48].

### 5.2. Promoting Heavy Metal Excretion

After being absorbed, heavy metals can accumulate in the soft tissues of the body, and the liver is the main target organ for heavy metal poisoning. Increased oxidative stress is the primary mechanism responsible for the liver toxicity after heavy metal poisoning and can induce aberrant inflammatory response. Thereafter, deregulated inflammation can enhance the production of reactive oxygen species, which can aggravate liver injury [68]. Excessive intake of heavy metals in the body can cause headache, dizziness, insomnia, etc., whereas the severe ones can lead to serious damage to organs such as brain, kidney, and skin [69]. The various strategies adopted to reduce the harmful effects of heavy metals by *L. plantarum* in vivo include promoting the excretion of heavy metals and reducing the content of heavy metals in the tissues (Figure 4). It can also protect intestinal barrier as well as inhibit intestinal absorption capacity of heavy metals by regulating oxidative stress to alleviate the toxicity of heavy metals in the tissues [70].

In another study, Dwyana Z confirmed that *L. plantarum* exhibited a significant effect in reducing the blood concentration of methylmercury [71]. Interestingly, results of in vivo experiments in mice have shown that *L. plantarum* CCFM8610 could effectively reduce the cadmium content in mice exposed to cadmium and promote the excretion of cadmium through feces [72]. Moreover, in combination with B. lactis, the detoxification rate of heavy metals was found to be as high as 90.7% when *L. plantarum* 33 ($1.0 \times 10^{10}$ CFU/kg, once a day) was given to lead-exposed rats for 8 weeks [34]. It was found to reduce the liver injury rats caused by lead exposure, as well as inhibit oxidative stress, inflammatory response, and activation of nuclear factor κB signaling in the liver tissues. In addition, exposure to *L. plantarum* WSJ-06 attenuated memory dysfunction and reduced inflammatory cytokine levels in both the serum and hippocampus caused by lead exposure. *L. plantarum* WSJ-06 attenuated the neurobehavioral impairment caused by accumulation of lead in the mice

through modulation of intestinal flora. It also partially restored the lead-induced dysbiosis of the intestinal microbiota. It increased the proportion of a few beneficial metabolites in the serum, such as arachidonic acid, tryptophan hydroxylase, serotonin, vitamin B12, trehalose and kynurenine. WSJ-06 also decreased the levels of harmful metabolites in the serum, such as lipopolysaccharides and L-kynurenine [73].

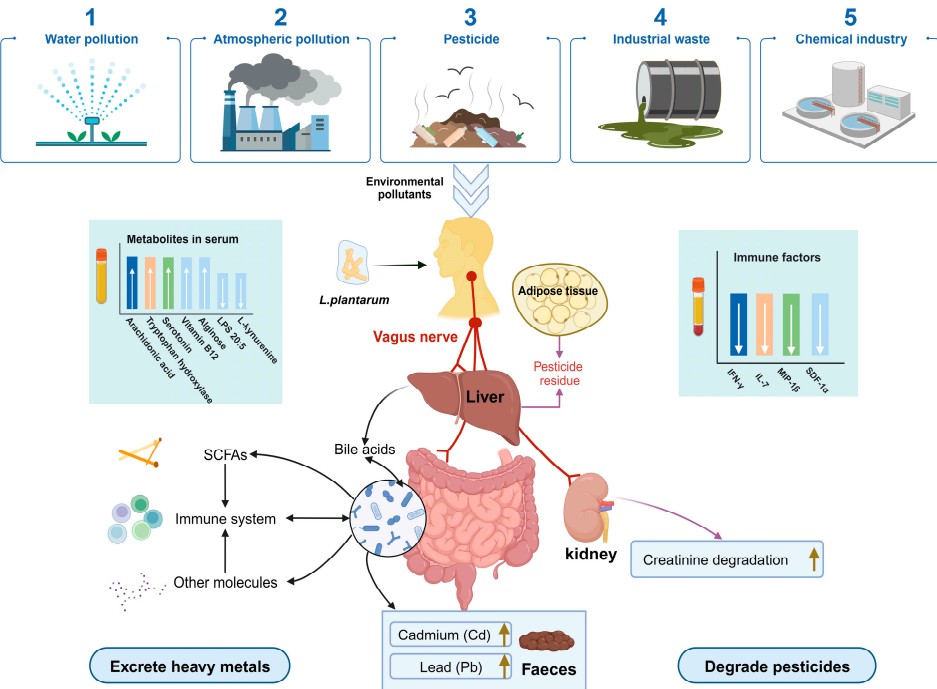

**Figure 4.** Protective effects of *L. plantarum* on adverse effects caused by environmental pollutants (pesticides and heavy metals) on the human body. (Black arrow) The mechanisms that can promote the excretion of heavy metals after ingestion by exogenous *L. plantarum*, including the regulation of the beneficial and harmful metabolites in serum; (purple arrow) Mechanisms contributing to the degradation of pesticides, including the regulation of levels of some immune factors in the blood [68,70].

### 5.3. Degradation of the Pesticides

Pesticides are a class of substances that can prevent, eliminate, or control pests, including chemical and biological pesticides. In the human body, organophosphorus pesticides are transported to the liver through the gastrointestinal tract and blood circulation. The liver is the main organ involved in pesticide metabolism, and the gastrointestinal tract is the main target of pesticide degradation. In addition, a number of prior studies have indicated that organochlorine pesticides (OCPs) can alter the relative abundance and composition of intestinal microbiota, especially in enhanced *Lactobacillus* with BSH activity. In addition, OCPs also affected bile acid composition, enhanced hydrophobicity, and decreased expression of genes regulating bile acid reabsorption in the terminal ileum, which led to a compensatory increase in the expression of the various genes involved in the synthesis of bile acids in the liver [74]. *L. plantarum* has also been shown to be effective in degrading the pesticide residues in the body (Figure 4) [75].

*L. plantarum* P9 can effectively alleviate phorate poisoning in rats, improve the degradation rate of pesticides in the intestine, can also maintain the internal balance of intestinal microbiota, and, thus, can reduce the abundance of pathogens in subjects with high pesticide exposure. In addition, P9 was also found to alleviate the disorders related to immune factors and cause a significant decrease in expression of INF-γ, IL-7, and SDF-1α (stromal cell-derived factor 1α) inflammatory factor level, and macrophage inflammatory protein (MIP-1 β), but upregulate the levels of creatinine degradation II pathways, which can facilitate increased excretion of the pesticide residues [30,76]. *L. plantarum* L-137 (50 mg/kg)

was found to protect Nile tilapia exposed to deltamethrin (15 μg/L) for 30 days. The blood biochemical parameters (creatinine, urea, bilirubin), liver enzymes (alkaline phosphatase, glutamic pyruvic transaminase, glutamic pyruvic transaminase), blood total protein, globulin, albumin, leucocyte, erythrocyte, hemoglobin, phagocytic index, phagocytic cells, and lysozyme activity were restored to the normal levels [77].

*5.4. Improving Body Movement Ability*

Exercise can cause significant alterations in the body and these changes include the impact of millions of microorganisms living in our gut. In addition, a two-way communication mechanism has been reported between intestinal flora and skeletal muscle [78]. Glucose is converted into lactate in the muscle, enters the liver through the blood circulation, and then is converted back to glucose in the liver through gluconeogenesis. The lactic acid produced in the muscle then can enter the intestinal cavity through the blood circulation. In the intestine, it acts as a potential carbon source of specific microorganisms, leading to a significant increase in *Veillonella* and producing the SCFAs (mainly propionate) (Figure 3), thereby leading to improvement in exercise performance [79]. Moreover, prior scientific research has shown that probiotics can be useful for athletes involved in long-term large exercise training and fierce competition prone to inflammatory response as well as acute upper respiratory tract infection as they can even promote muscle recovery and improve the body's overall athletic ability [80,81].

Several preclinical studies have shown that after feeding *L. plantarum* TWK10 ($1.03 \times 10^9$ CFU/kg/d) for 6 weeks, male mice displayed stronger forelimb grip, more muscle mass, and longer swimming time. It could reduce the accumulation of lactic acid induced by exercise, as well as the concentration of ammonia and creatine kinase in the serum, and increase the utilization rate of glucose. Moreover, *L. plantarum* TWK10 could also increase the number of gastrocnemius muscle type I muscle fibers (represented by the white muscle fibers) [35]. Based on the positive results of these animal trials, further relevant clinical trials were carried out. In a randomized, double-blind, placebo-controlled trial involving 16 adults over 20 years of age, researchers found that the supplementation with *L. plantarum* TWK10 ($1 \times 10^{11}$ CFU/d) could markedly improve exercise tolerance [82]. Another double-blind placebo-controlled trial involving 54 subjects aged 20–30 showed that *L. plantarum* TWK10 could significantly improve exercise performance in a dose-dependent manner. Furthermore, compared with the placebo group, the exercise performance of the group supplemented with *L. plantarum* TWK10 was significantly increased by 36.76% [83]. Moreover, another study analyzed the effectiveness of human-origin *L. plantarum* PL-02 in improving muscle mass, exercise performance and overcoming fatigue. Interestingly, supplementation with PL-02 for 4 weeks significantly increased the muscle mass, muscle strength, and endurance performance, as well as liver and muscle glycogen stores. In addition, *L. plantarum* PL-02 was observed to reduce the post-exercise levels of lactate, blood urea nitrogen (BUN), ammonia, and creatine kinase (CK) ($p < 0.05$) [84]. *L. plantarum* DR7 was administered for 12 weeks (9 $\log_{10}$ CFU/d, equated with $1 \times 10^9$ CFU/d) in another clinical study. It was observed that administration of DR7 markedly reduced the duration of nasal symptoms and the frequency of URTI after 4 as well as 12 weeks in comparison to the placebo. Interestingly, DR7 treatment also suppressed the levels of plasma proinflammatory cytokines (IFN-γ, TNF-α) in middle-aged adults (30 to 60 yr old) but enhanced that of anti-inflammatory cytokines (IL-4, IL-10) in young adults [26].

## 6. *L. plantarum* with Immunity

*6.1. Regulation of Immunity*

Immunity refers to the ability of the body to resist external invasion and maintain the stability of the internal environment. Low or ultra-long immunity can cause several harmful effects on the body, such as allergic reactions and autoimmune diseases. *L. plantarum* can substantially increase the levels of IgA, IgG, and IgM in both the mucosal surface and serum through affecting bacterial cell or wall components, and can enhance humoral immunity

(Figure 2). It has also been reported to promote the proliferation of T lymphocytes as well as B lymphocytes and strengthen cellular immunity. It can enhance the activity of mononuclear phagocytes (monocytes and macrophages), stimulate the secretion of reactive oxygen species, lysosome enzymes and mononuclear factors, and thereby stimulate the non-specific immune response [85].

*L. plantarum* DR7 alleviated the symptoms of upper respiratory tract infection by improving the various inflammatory parameters and enhancing immune regulation characteristics in a randomized, double-blind, and placebo-controlled trial of 109 adults [26]. In addition, *L. plantarum* P-8 could effectively regulate the immune status in the humans and promote elevated levels of immunoglobulin SIgA in middle-aged as well as elderly populations (14 days of consumption). The administration of *L. plantarum* IS-10506 was found to be effective in alleviating atopic dermatitis symptoms in adults, owing to its immunomodulatory effects. The SCORAD (SCORing Atopic Dermatitis) score was significantly lower in the *L. plantarum* IS-10506 in comparison to the placebo group after 8 weeks of continuous capsule administration (containing *L. plantarum* IS-10506, $10^{10}$ CFU/d). The IL-4 and IL-17 levels were also significantly lower in the *L. plantarum* group than the placebo group. However, IFN-$\gamma$ and Foxp3+ levels were significantly higher in the *L. plantarum* IS-10506 than the placebo group [86].

Administration of drinking water containing *L. plantarum* P-8 ($2 \times 10^6$ CFU/mL) for 42 days to the broilers could lead to improved levels of IgG and intestinal endocrine IgA in the blood, promote the development of broiler mucosal immune system, and enhance the immune function of intestinal-related lymphoid tissue [47]. In addition, another study confirmed that $10^8$ CFU/g supplementation of *L. plantarum* VSG3 (3% of body weight per day) in fish diets could enhance their growth, immunity, as well as anti-streptococcal infection ability, and exhibited the ability to improve immunity in the different feeding environments [25]. *L. plantarum* 0111 could protect against influenza virus by modulating intestinal microbial-mediated immune responses. Pretreatment of *L. plantarum* 0111 was observed to effectively improve the survival rate and weight loss of mice. It could also decrease the levels of inflammatory cytokines in the lung and bronchoalveolar lavage fluid, as well as reduce the degree of lung and intestinal injury [87].

*6.2. Effect on Absorption of the Trace Elements*

The trace elements mainly include vitamins (vitamin A, vitamin B6, vitamin B12, vitamin C, vitamin D, vitamin E, and folic acid), zinc, iron, selenium, magnesium, and copper; $\omega$-3 fatty acid eicosatetraenoic acid and docosahexaenoic acid (DHA), etc. These elements play an important role in maintaining the normal health of the human body. Micronutrient deficiency decreases immunity, and even their low utilization can adversely affect it. The body's resistance to infection can negatively affect immune function, thereby increasing the disease hazard [88,89]. The addition of probiotics to food has gradually emerged as an effective way to promote the absorption of trace elements. Fermented food can be obtained by reducing pH-value, activating phytase, producing organic acids, or modulating absorption of non-heme iron with live LAB, which was found to significantly increase the absorption of some trace elements in the human body [90].

A few studies have also shown that *L. plantarum* 299v can effectively promote the absorption of trace elements of iron. In a study, healthy women of childbearing age were given fruit drinks for 4 consecutive days (200 mL/d) (containing *L. plantarum* 299v, $10^9$ CFU or $10^{10}$ CFU). The results showed that the average iron absorption of the drink containing $10^9$ CFU *L. plantarum* 299v or $10^{10}$ CFU *L. plantarum* 299v was significantly higher than that of the control drink. Moreover, intake of probiotics can increase iron absorption by approximately 50% from a fruit drink having an already relatively high iron bioavailability [91]. After that, in a single-blind placebo-controlled dual-isotope (59Fe and 55Fe) design, the iron uptake of a light breakfast diet containing $10^{10}$ CFU freeze-dried *L. plantarum* 299v capsules was studied in the healthy women of childbearing age [92]. Freeze-dried *L. plantarum* 299v was found to effectively enhance iron absorption when

combined with high iron bioavailability diet. Similarly, the effect of daily supplementation of *L. plantarum* 299v ($10^{10}$ CFU) and 20 mg iron (ferrous fumarate) to low iron reserve (ferritin < 30 g/L) in the healthy anemia-free female athletes was monitored. It was found that 4 weeks later, the intake of iron-containing *L. plantarum* 299v could increase ferritin levels (13.6 vs. 8.2 g/L), which was more than the intake of iron alone [93]. After 12 weeks, in comparison with the single intake of iron, the average content of reticulocyte hemoglobin after intake of *L. plantarum* 299v containing iron was observed to increase significantly (1.5 vs. 0.82 pg).

## 7. Effect of *L. plantarum* on Nervous System

In recent years, the gut microbiota as a component of the gut–brain axis has been reported to play an important role in neuroscience and psychiatry. *L. plantarum* has been shown to improve mood, synaptic ability, depression, and cognitive ability, especially in some memory disorder organisms, and exhibit beneficial protective effects (Figure 2) [94].

In a previous study, our team confirmed, through a randomized, double-blind, and placebo-controlled trial lasting for 12 weeks, that intake of $2 \times 10^{10}$ CFU/d of *L. plantarum* P-8 could alleviate the stress and anxiety state of a specific subjects (all subjects fulfilled the criteria of moderate stress upon diagnosis using the PSS-10 questionnaire). The possible mechanism was attributed to the reduction in stress and anxiety symptoms through anti-inflammatory effects, which could enhance both memory and cognitive ability [49]. Furthermore, genome assembly analysis of fecal samples showed that the contents of *B. adolescentis*, *B. longum*, and *Fecalibacterium Prausnitzii* increased significantly after *L. plantarum* P-8 intervention. However, the contents of *Roseburia faecis* and *Fusicatenibacter saccharivorans* in feces were significantly decreased, thus indicating that *L. plantarum* P-8 can promote the production and secretion of some key neurotransmitters as well as neuroactivators in the intestine by regulating intestinal flora. It could also directly as well as indirectly modulate the intestinal–brain axis by affecting the vagus nerve, cytokines, and microbial metabolites, thereby alleviating adult stress and anxiety symptoms [95].

In addition, other studies have shown that after administration of *L. plantarum* C29 fermented soybean (DW2009) to MCI (Mild Cognitive Impairment) patients for 12 weeks, the comprehensive cognitive function score of the *L. plantarum* group was significantly improved, especially in terms of attention. The findings of a randomized, double-blind, and parallel-controlled trial lasting for 8 weeks indicated that administration of *L. plantarum* 299v ($20 \times 10^9$ CFU/d) to patients with MDD (major depressive disorder) could effectively reduce the concentration of canine uric acid and reduce the plasma KYN concentration through a variety of assumptions, thereby improving cognitive function in MDD patients [96]. In a double-blind trial, participants took two capsules of either *L. plantarum* PS128 or a placebo after dinner for 30 days. Compared to the control group, the *L. plantarum* PS128 group showed significant decreases in Beck Depression Inventory-II scores, fatigue levels, brainwave activity, and awakenings during the deep sleep stage. Overall, daily consumption of *L. plantarum* PS128 as a dietary supplement may improve the depressive symptoms and sleep quality of insomniacs [38]. In similar experiments, *L. plantarum* PS128 also showed a significant improvement in autism spectrum disorder [39] and MDD [40].

In another study, *L. plantarum* 69-2 was combined with galacto-oligosaccharides (GOS) and supplemented in a d-galactose (d-gal)-induced neurodegeneration and memory impairment mouse model. *L. plantarum*-GOS supplementation markedly inhibited d-gal-induced oxidative stress and increased the nuclear factor erythroid 2-related factor 2 (Nrf2) levels in the brain [97].

## 8. *L. plantarum* with Oral and Skin

### 8.1. Promoting Oral Health

Oral microorganisms are diverse, and mainly consist of *Streptococcus*, *Veillonella*, as well as *Proteobacteria*. The diversity of species, genes, and metabolism in oral microflora

can reflect the health and disease status of the mouth and entire body. Probiotics have been used to treat various oral diseases, as they can significantly improve oral health, reduce the occurrence of dental caries, paradentitis (*Porphyromonas gingivalis*, *Actinobacillus actinomycetemcomitans*, *Fusobacterium nucleatum*), and halitosis (Figure 5). They have been found to be effective in significantly reducing the number of cariogenic bacteria (Streptococcus mutans) in the saliva and oral biofilm, thereby promoting oral health maintenance in patients receiving fixed oral therapy [98].

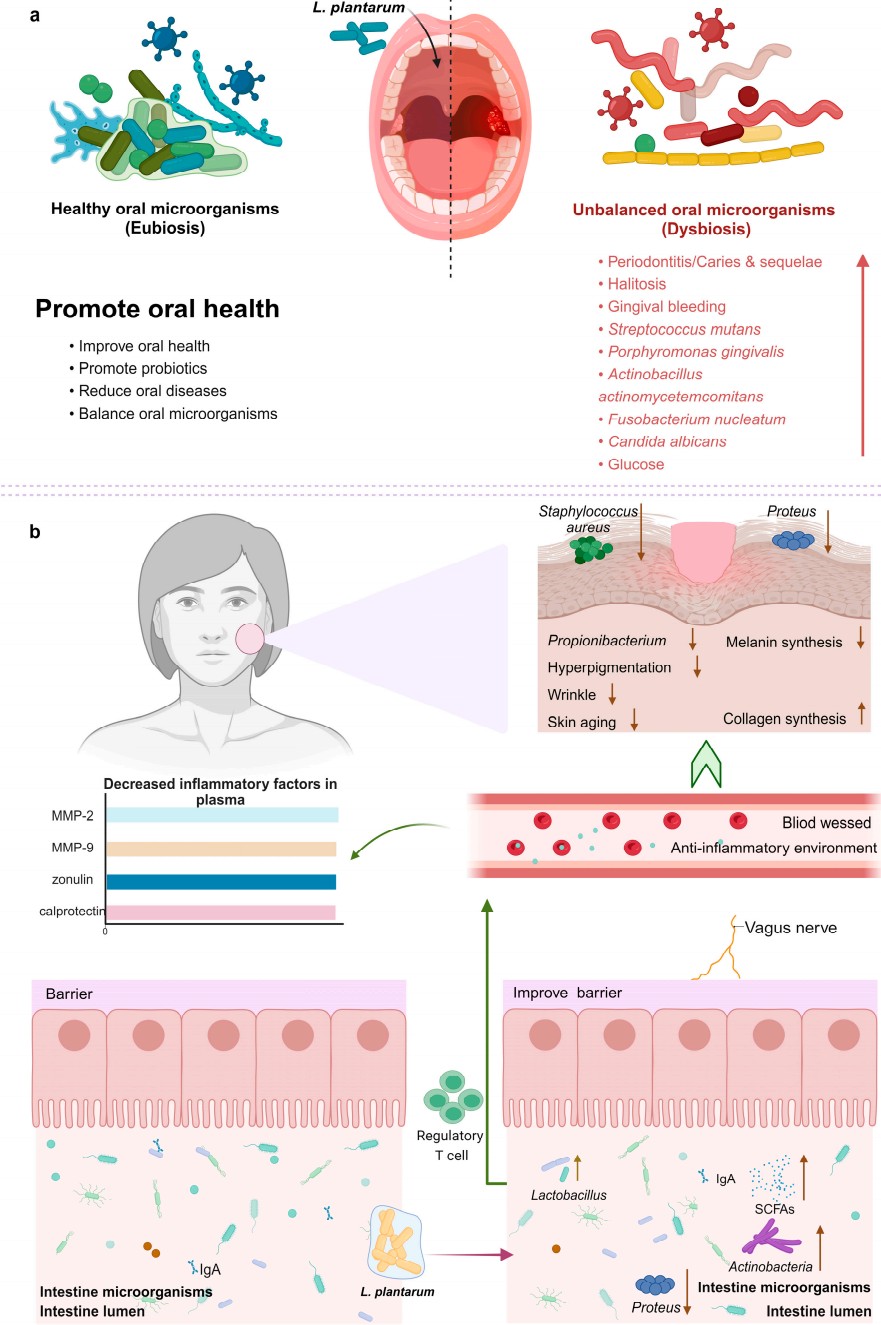

**Figure 5.** Interaction mechanism of *L. plantarum* with human mouth and skin. (**a**) Oral mechanism of action of exogenous *L. plantarum* after ingestion, based on comparison of healthy oral flora and non-healthy oral flora. (**b**) Mechanisms of action on the skin, combined with skin–gut axis effects, regulation of the intestinal as well as skin microflora, and levels of inflammatory factors in the plasma [44,50,99].

From the perspective of oral microecology, probiotics can play a beneficial role in both the prevention and treatment of decayed teeth in children, and periodontitis disease in adults. *L. plantarum* CCFM8724 by reducing the number of oral pathogens and altering the oral microbiota in children with dental caries. It also reduced the amount of *Candida albicans* in the saliva of children with decayed teeth ($p < 0.01$) [45]. *Lactobacillus brevis* and *L. plantarum* strains were applied as gels to periodontal pockets in a double-blind, randomized, placebo-controlled experiment, and then used as potential adjuvants for ingot preparation, cleaning, and root surface leveling. The patients were treated with cleansing and root planing for 7 days (twice a day), followed by probiotics gel and ingot or placebo treatment. The number of diseased sites was reassessed after 3 months, and it was found that periodontitis patients benefited significantly from the adjuvant use of probiotics containing *L. brevis* and *L. plantarum* in reducing gingival bleeding [99].

*8.2. Improving Skin Flora*

Human skin represents one of the largest organs of the body and is home to a highly diverse microbial community, dominated by actinomycetes, Proteobacteria, and Firmicutes, whereas fungi account for 4% of the skin flora. The skin microbiome varies with the parts of the body, and this is related to temperature, humidity, pH-value, and other external factors. Each person has a unique and relatively stable skin microbiome [100]. There are several common essential features found in the skin and intestine, and their normal functions are extremely crucial for maintaining the balance and survival of the whole organism [101]. It is recognized that the first target of probiotics is the intestinal flora, and *L. plantarum* has been reported to promote skin health by improving intestinal conditions, which is the intestine–skin axis (Figure 5), to achieve its functional role in the skin [50].

*L. plantarum* has exhibited the potential to delay skin aging and inhibit wrinkle formation as well as hyperpigmentation in clinical trials [102]. Moreover, local application studies on the female subjects have showed that *L. plantarum* GMNL6 can reduce the proportion of *Propionibacterium*, and that heat-killed *L. plantarum* GMNL6 had the ability to reduce erythema and melanin [46]. Information from clinical observation during the use of ointment for the external face (contain *L. plantarum* GMNL6) by people displayed that the syndromes of skin moisture, skin color, spots, wrinkles, UV spots, as well as porphyrins were markedly improved, and that microbiomes were affected. GMNL6 enhanced collagen synthesis and the gene expression of the *serine palmitoyltransferase small subunit A*. In addition, GMNL6 reduced melanin synthesis, the biofilm of *Staphylococcus aureus*, and the proliferation of *Cutibacterium acnes*. In addition, administration of *L. plantarum* CJLP55 ($1.0 \times 10^{10}$ CFU/d) for 12 weeks remedied acne in patients with mild-to-moderate facial acne in a double-blind, placebo, randomized controlled trial. *L. plantarum* CJLP55 was found to effectively improve the count and grading of acne lesions, reduce the triglyceride (TG) of sebum, increase hydration, maintain the main neuramide-2 of epidermal lipid barrier, and reduce the prevalence of Proteus [44]. Interestingly, heat-inactivated *L. plantarum* L-137 markedly improved the dry skin and increased skin satisfaction, which can affect both skin water content and transcutaneous water loss [103]. Additionally, *L. plantarum* HY7714 significantly ameliorated inflammation by reducing matrix metallopeptidases (MMP-2 and MMP-9), zonulin, and calprotectin expression in the plasma, all of which are involved in regulation of skin and intestinal permeability [50].

**9. Conclusions**

As a candidate for probiotics, *L. plantarum* has demonstrated multifaceted probiotic functions. A number of preclinical trials, in vitro simulations, and clinical experiments using *L. plantarum* have established that it can exhibit a variety of probiotic functions through diverse mechanisms that have led to its widespread applications. Several distinct strains or recombinants of *L. plantarum* can also be used as vaccine carriers and in major industries such as food medicine. In short, *L. plantarum* has still not been used commonly in everyday life for health improvement, primarily because ordinary consumers are unaware

of its various health promoting effects. In other words, *L. plantarum* is only employed at small-scale level for evaluation and comparison. It has not yet been recognized as a common product to be used for daily necessities. To further refine its practical applications, its role needs to be verified at the individual level, and more accurate, large-scale, as well as comprehensive evaluation is needed to provide more theoretical basis for the development of functional probiotic LAB products. Moreover, to facilitate its application in various industries, we should aim to enrich the dosage forms of *L. plantarum*, pay more attention to the dose–response relationship, validate its potential for recent health-related issues reported by consumers, and use it effectively in medical, sanitary, and biological products for both animals and plants.

**Author Contributions:** Y.H., conceptualization, software, validation, visualization, writing—original draft preparation; J.L., validation, investigation; J.W., writing—review and editing, funding acquisition, supervision; Y.C., writing—review and editing, funding acquisition, supervision. All authors have read and agreed to the published version of the manuscript.

**Funding:** This study was funded by the National Natural Science Foundation of China (32260572), and the National Natural Science Foundation of China (31972053).

**Institutional Review Board Statement:** Not applicable.

**Informed Consent Statement:** Not applicable.

**Data Availability Statement:** Not applicable.

**Conflicts of Interest:** The author declares no conflicts of interest.

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
