# Peer review of "Mechanisms of Health Improvement by Lactiplantibacillus plantarum Based on Animal and Human Trials: A Review"

_fermentation, doi:10.3390/fermentation10020073_

Round 1

Reviewer 1 Report

Comments and Suggestions for Authors

Dear Authors

The manuscript entitled “Mechanisms of health improvement by Lactiplantibacillus plantarum based on preclinical and human trials: A review” written by Y. Hao et al. is reviewed the both preclinical and human studies related to the health promoting effects of L. plantarum. This article is well summarized the mechanisms related direct or indirect health promoting effects by L. plantarum as the probiotics. However, some improvements are required to publish in this journal. Please refer to the following comments for revision.

Comments:

1)      Please reconsider the keywords because they have multiple terms as on keyword.

2)      It is difficult to identify the characters in all figures (e.g. very small letters, the color of characters similar with background color).

3)      Page 5 of 18: In the line 1, “Sutterella, Pediococcus, Proteus, Parabacterioides, Prevotella” and “Desulfovibrio” should be corrected to italics.

4)      Please reconsider the caption of Fig. 1 to the appropriate.

5)      Are the graphs used in the figures cited data? Is so, please cite their references in the figure legends.

6)      Please correct the mixed letter case large and small in the title and column titles in Table 1.

7)      Please standardize the format of the reference list.

8)      Please make accurate the list of references (e.g. name of authors, article pages, journal title, etc.).

9)      In the subtitle of Figure 4, all of characters described as “L. plantarum” are large letters. Please correct it.

10)   Page 10 of 18: In the line 4, what is meaning of “9 log CFU/d”?

11)   In the subtitle of Figure 5, all characters described as “L. plantarum” are large letters. Please correct it.

12)   In Figure 3, the term of “Brain” should be horizontally. Please reconsider the color of characters in figure because the texts are similar in color to the background illustration.

13)   Please reconsider the color of them since they cannot distinguish the color of characters and the background illustration.

14)   The subtitle of “7.1. Enhancement of memory” should be removed.

15)   Page 14 of 18: Subheadline of “9. Conclusion” should be inserted new line.

Reviewer 2 Report

Comments and Suggestions for Authors

The authors have reviewed both animal and human studies related to the health promoting effects of Lactiplantibacillus plantarum. In addition, the authors providing a theoretical basis for the development of various beneficial applications of L. plantarum.

The following question arises.

Title: “based on preclinical and human trials”

It is recommended that the title be changed to “based on animal and human trials”.

Section 2:

1. “with required pH-in range of 6 to 7”

The authors need to verify its accuracy.

Section 3:

1. dairy product[11,12] s

The authors need to verify its accuracy.

Table 1:

The authors shall summary the functions of all the L. plantarum strains present in the review.

Section 5.1

1. “Here we have discussed the potential of probiotics in affecting hyper-lipidemia in detail [30]”.

The reference [30] is incorrect.

2. “Woo J Y et al. have reported that oral administration of L. plantarum C29 (1×1010CFU/mouse, p.o.) to D-galactose-induced aging mice for 5 weeks, significantly decreased the expression of senescence marker p16 and inflammatory markers p-p65, pFOXO3 cyclooxygenase (COX)-2 and inducible nitric oxide synthase. It also inhibited the levels of tumor necrosis factor-α and arginine enzyme II, and decreased the expression of

M2 markers IL-10, arginine enzyme I and CD206[32].”

The experiment is about ameliorates memory impairment and inflammaging in a d-galactose-induced accelerated aging mouse model. It is unrelated to reducing cholesterol.

Section 5.2

1. Alginose shall be changed to trehalose according to the reference 44.

2. What is LPS 20:5?

Section 5.3

1. The full name of SDF-1α shall be written.

Section 5.4

1. “significant increase of veronella”

The authors need to verify its accuracy.

Section 6.1

1. What is the SCORAD score? The full name shall be written.

Section 7.1

1. What are the MCI patients and MDD patients? The full name shall be written.

2. The authors shall discuss the psychobiotic Lactobacillus plantarum PS128. The psychophysiological effects of the strain were tested in animal and human trials.

Section 8.1

1. “It also reduced the amount of glucose and Candida albicans in the saliva”

What is the importance of glucose in the experiment of reference [75]?

Section 8.2

1. “L. plantarum GMNL6 (GMNL6) can reduce the proportion of Propionibacterium in erythema and face[81]”

It is misleading. In reference [81], L. plantarum GMNL6 had the ability to reduce the erythema and melanin. The authors need to verify its accuracy.

2. “L. plantarum CJLP55 was found o effectively improve the count and grading of acne lesions”

The authors need to verify its accuracy.

3. “Interestingly, heat-inactivated L. plantarum L-13”

L-13 shall be corrected to L-137

Round 2

Reviewer 2 Report

Comments and Suggestions for Authors

All errors have been corrected, and the article is acceptable.

Author Response

Dear editor,
We have explained all the questions raised in the email. The problems and explanations are as follows :
1. We noticed that the email address of Dr. Jianli Li ( imtt6878 @ sina.com ) is undeliverable. Please check and provide the correct one.
AU : We will re-provide an email about Dr. Li Jianjian that can receive emails, [email protected].

2. We noticed that there is one citation in Figures 1, 2 and 3 legend. We are wondering whether this figure is involved with copyright issues. If the answer is yes, we need you to provide copyright permission. If the answer is no, please provide the PDF of the cited article so that we can confirm it isn’t involved with copyright permission. We intend to avoid some dissension appearing between you and the copyright owner after your paper is published.
AU: First, Figure 1 contains three pictures, namely P-8 colony morphology, cell morphology under microscope and electron microscope. Among them, the cell morphology under microscope, we have uploaded the relevant PDF files in the form of attachments(does not involved with copyright issues); the colony morphology and the electron microscopy were recorded during the past experiment (P-8 is a strain of our team and does not involved with copyright issues) ;
Secondly, the role of references in each legend is to refer to the metabolic process in the form of its text. I redraw the metabolic pathways as shown in the pictures according to the text (does not involved with copyright issues).
All problems do not need to be modified and updated in the manuscript. The final manuscript has been submitted.
If you have any questions, please feel free to contact us.

Sincerely,
Jicheng Wang
